TECHNICAL RELEASE

# SMARTdenovo: a *de novo* assembler using long noisy reads

Hailin Liu[1,†], Shigang Wu[1,†], Alun Li[1,†] and Jue Ruan[1,*]

1 Guangdong Laboratory for Lingnan Modern Agriculture, Genome Analysis Laboratory of the Ministry of Agriculture, Agricultural Genomics Institute at Shenzhen, Chinese Academy of Agricultural Sciences, Shenzhen 518120, China

## ABSTRACT

Long-read single-molecule sequencing has revolutionized *de novo* genome assembly and enabled the automated reconstruction of reference-quality genomes. It has also been widely used to study structural variants, phase haplotypes and more. Here, we introduce the assembler SMARTdenovo, a single-molecule sequencing (SMS) assembler that follows the overlap-layout-consensus (OLC) paradigm. SMARTdenovo (RRID: SCR_017622) was designed to be a rapid assembler, which, unlike contemporaneous SMS assemblers, does not require highly accurate raw reads for error correction. It has performed well in the evaluation of congeneric assemblers and has been successfully users for various assembly projects. It is compatible with Canu for assembling high-quality genomes, and several of the assembly strategies in this program have been incorporated into subsequent popular assemblers. The assembler has been in use since 2015; here we provide information on the development of SMARTdenovo and how to implement its algorithms into current projects.

**Subjects** Genetics and Genomics, Software and Workflows, Bioinformatics

**Submitted:** 19 November 2020

* Corresponding author. E-mail: ruanjue@caas.cn

† Contributed equally.

Preprint submitted at https://doi.org/10.20944/preprints202009.0207.v1

## INTRODUCTION

The development of high-throughput sequencing provides the means to deliver fast, inexpensive, and accurate information for assembling whole genomes. As a result, there has been rapid growth in the number of whole-genome sequencing projects [1–3]. Single-molecule sequencing (SMS) technologies, such as Pacific Biosciences (PacBio) and Oxford Nanopore, which generate sequencing reads of >20 kb in length, are now widely used in whole genome projects. These long reads are advantageous because they span polymorphic regions, repeats, and transposable elements, and because they provide long-range information for assemblies that are usually too complex to be resolved by short reads alone.

The huge demand for long-range DNA sequencing and mapping technologies has catalysed a renaissance of the development of high-quality SMS assemblers, such as PBcR [4, 5], Falcon (RRID: SCR_016089) [6], Canu (RRID: SCR_015880) [7], Miniasm [8], Ra [9], Wtdbg2 (RRID: SCR_017225) [10], Flye (RRID: SCR_017016) [11], Shasta [12], and ABruijn [13]. In fact, highly available SMS assemblers have always been essential for improving the quality of genome assemblies.

SMARTdenovo (RRID: SCR_017622) is a long-read SMS assembler that follows the overlap-layout-consensus (OLC) paradigm. It assembles genomes following four steps: overlapping, trimming, layout, and consensus. The source code for SMARTdenovo was released in GitHub in 2015 [14]. Assessment by others has shown that it performs well

compared with other congeneric assemblers [15], and it has been widely used for generating highly accurate contigs in many genome assemblies [16–18]. For datasets from both PacBio and Oxford Nanopore, such as 20–30× 2D reads of different varieties of yeast, SMARTdenovo assembled more accurate and more highly contiguous sequences than other assemblers [15, 19]. SMARTdenovo was also used successfully for datasets from the wild tomato *Solanum pennellii* (~1.2 Gb) and *Sorghum bicolor* (~732 Mb) using Oxford Nanopore reads [18, 20], and for the long-read datasets for *Taraxacum kok-saghyz* (~1.04 Gb) and the woody plant, *Rhizophora apiculata* (~274 Mb) using PacBio RSII reads [16, 21]. Here, we explain how we developed SMARTdenovo and provide use cases to show its ability.

## IMPLEMENTATION

### The SMARTdenovo algorithm

SMARTdenovo uses four steps for assembly: overlapping, trimming, layout, and consensus. We used homopolymer-compressed (HPC) *k*-mers for seed-indexing and identifying collinear seeds. HPCs have been widely adopted in Minimap2, Wtdbg2 and Shasta [10, 12, 22]. We then trimmed low-quality regions and chimeric reads based on the overlapping reads. We applied the Best-Overlap-Graph [23] to generate the layout of the reads and the PBDAG-Con algorithm [24] to generate a consensus.

#### *Overlapper*

The algorithm for alignment follows a typical seed-chain-align procedure that is used by most full-genome aligners. In this step, we constructed each read by contracting homopolymer reads to a single base, called HPC strings. An HPC *k*-mer, a 1000-bp substring of an HPC string, was treated as a seed ("wtzmo -k 16 by default"). For HPC-based *k*-mer-indexing, we scanned all the reads and counted *k*-mers in a hashtable with a 64-bit key to store a *k*-mer, and a 64-bit value to store its count. If a *k*-mer was present more than 500 times, it was filtered out ("wtzmo -K 500"). A seed array was also created and filled with the "read_id" and the orientation of the remaining seeds. To manage the cost of memory for *k*-mer-indexing, we used two different parameters: (1) we kept only the index with smaller values between the *k*-mer and its reverse complement; (2) we randomly selected *k*-mers according to the hash code (one quarter was set as the default). All queried *k*-mers were indexed against the hashtable and seed array to identify candidate reads. We sorted the seeds by the "read_id" and "strand" and calculated the coverage length of the overlaps. If the coverage was longer than 300 bp ("wtzmo -d 300"), the candidate was kept. The top 500 candidates were chosen for each query ("wtzmo –A 500").

To refine the collinearity relationship between query and candidate, we further built a similar but shorter HPC-based index called a "z-mer" on queries for seed chaining. Each *z*-mer was recorded with its offset and strand. Pairs of matched seeds between candidate and query were obtained. However, because the high rate of insertions and deletions (indels) made the distance between two nearby *z*-mers highly variable, wtzmo was used to identify the synteny between query and candidate using sliding windows ("wtzmo –y 800") on query instead of using the whole overlapping region. We first filtered *z*-mer windows using a minimum match length of 200 bp. We then developed a scoring algorithm to filter excessive matches. The adjacent matched pairs of seeds ($p_i$, $p_{i+1}$) were scored based on the relationship of *z*-mers on candidates: $S_{i+1} = S_i + L_{i+1} - Distance_{i-(i+1)}$, *S*, *L*, *D* represented the sort of seeds, length of seeds, and the distance between the adjacent seeds, respectively. If



$S_{i+1}$ was larger than $L_{i+1}$, pairs of $p_i$, $p_{i+1}$ were considered to be a syntenic block, and the block would be extended until $S_j$ was smaller than $L_j$. At that point, the block ended, $S_j$ recovered to $L_j$ and the next round of syntenic block identification began. The $z$-mer-block containing the highest value of S was the block chosen as the best syntenic overlap between the two windows. If the coverage was longer than 100 bp, the matched windows were retained. Seed windows were also scored using the same method as above for searching the best colinear pairs. Finally, candidates with the best colinear window-block that covered more than 300 bp were retained for pairwise alignment.

To speed calculation time and reduce unnecessary computation, we developed a weighting algorithm that negatively ranked a repetitive region based on its depth in the alignment process. "Wtzmo –q 100" was set so that the weights ranged between 1 and 0 if the depth ranged from 10–100. Large numbers of false candidates containing similar repeats were eliminated with queries. The pairwise alignment procedure based on the collinearity relationship was split into four steps: (1) first, the bases of the $z$-mers were decompressed and gaps were added between the matched $z$-mers; (2) global alignment was conducted between two adjacent $z$-mers within a $z$-mer window; (3) the two adjacent $z$-mer windows were aligned using the banded global Smith–Waterman algorithm, with a dynamic bandwidth that increased according to the length of the gaps ("wtzmo –w 50 ~ 3200"); (4) the two ends were extended by global alignment with the band width set at 800 bp (Figure 1).

### Trimming
Wtclp trimmed or discarded reads to a maximum total length of the valid overlaps. It took one read as a reference, and tiled all reads containing overlaps. A functional model, "call_legal_overlaps_wtclp", calculated the length of valid overlaps. First, it clipped the ends that had high error rates. Then, it detected chimeras and trimmed them according to their depths. Structures containing partially aligned reads were called "spurs". We counted the depth of reads crossing the "spurs" as "$m$" and counted the number of reads with partial alignments as "$n$". If the $m$ of a read was less than half of the average depth, or if $n$ was larger than the average depth, or if $n$ was larger than "$m/2$", the read was considered chimeric and discarded. Otherwise, it was considered to be a sequencing error and the maximum region of reads was retained. Errors in the structure were corrected based on the graph. If a single read connected two subgraphs, it was considered a chimera. We then used wtclp to check for any alternative path formed by valid overlaps of tiled reads.

### Layout
Wtlay was used to achieve the Best-Overlap-Graph [23] to generate a layout of reads following the OLC paradigm. In general, if an overlap was not end-to-end, leaving $n$ (no greater than 100) bp unaligned, it would be treated as true ("wtlay –w 100"). Owing to the high indel rate, wtlay identified the best overlap with an alignment score ≥ 0.95, instead of picking out the longest one ("wtlay –r 0.95"). Using this process would keep bubbles from being merged, and instead find one appropriate path. The wtlay script also filtered out each unitig sharing more than 40% identity with another unitig to avoid islands. The output included uncorrected unitigs and all the parameters needed by the consensus caller.

### Consensus
We used the wtcns command to implement the PBDAG-Con algorithm described in HGAP to generate consensus [24]. Because an alignment algorithm is integrated into wtcns, it



**Figure 1.** The algorithm for processing overlaps. (a) *k*-mer indexing and identification of candidate reads based on overlap coverage. LR, long reads. (b) Seed chaining based on sliding windows. (c) Weighting algorithm negatively ranking the repetitive regions according to their depth. R, repeat; red blocks represent reads and candidates represent the repeat region; and these bases of high depth would be marked with lower weight. (d) Pairwise alignment with four steps: (1) decompress the matched *z*-mers; (2) global alignment between two adjacent matched *z*-mers within a *z*-mer window; (3) global dynamic SW-alignment between the two adjacent paired *z*-mer windows; (4) extend the two ends by global alignment.

required no other alignment tools. Wtcns took the layout file as input and output the consensus sequences in fasta format.

## RESULTS

## Assembling the genome of the fruit fly and evaluating accuracy of the assembly

We benchmarked SMARTdenovo against the SMS assemblers Flye, Canu, and Ra using the dataset of the fruit fly, *Drosophila melanogaster*, and calculated the accuracy of the assembly by aligning it with the reference genome. A total of 29.3 Gb PacBio reads from the National Center for Biotechnology Information Sequence Read Archive database (SRX499318) [25] was assembled with the command line "smartdenovo.pl -c 1 -t 16 reads.fa > wtasm.mak && make -f wtasm.mak". The length of the genome sequence was 146 Mb,



**Table 1.** Evaluation of long-read assemblies on the fruit fly genome (PacBio datasets).

| Parameter | SMS assembler | | | |
|---|---|---|---|---|
| | Canu | Flye | Ra | SMARTdenovo |
| Total length, Mb | 161.98 | 137.02 | 139.27 | 146.29 |
| Count of contigs | 633 | 970 | 424 | 242 |
| Total length (≥50,000 bp), Mb | 151.43 | 131.37 | 135.14 | 143.15 |
| Largest contig, Mb | 25.87 | 25.74 | 5.82 | 23.29 |
| N50, Mb | 21.40 | 9.57 | 1.02 | 11.59 |
| L50, *n* | 4 | 4 | 37 | 4 |
| Misassemblies, *n* | 3,296 | 783 | 242 | 1,382 |
| Mismatches per 100 kb, *n* | 158.82 | 36.93 | 63.60 | 103.44 |
| CPU hours | 2810.14 | 348.47 | 226.28 | 830.95 |
| peak MEM, Gb | 15.70 | 134.66 | 92.50 | 24.02 |

**Table 2.** Comparison of different assemblers on the wild tomato genome (Oxford Nanopore datasets).

| Parameter | SMS assembler | | | |
|---|---|---|---|---|
| | Canu | Flye | Ra | SMARTdenovo |
| Total length, Mb | 801.62 | 1265.09 | 815.60 | 902.99 |
| Count of contigs, *n* | 14,286 | 10,323 | 6490 | 4395 |
| Total length (≥50,000 bp), Mb | 606.49 | 1,070.26 | 767.88 | 860.21 |
| Largest contig, Mb | 2.25 | 4.54 | 2.98 | 3.34 |
| N50, kb | 114.88 | 429.25 | 161.11 | 399.45 |
| L50, *n* | 1728 | 720 | 1,292 | 633 |
| Misassemblies, *n* | 97 | 3539 | 1096 | 136 |
| Mismatches per 100 kb, *n* | 881.26 | 2656.12 | 2250.65 | 931.47 |
| CPU hours | 11,885.30 | 723.91 | 507.97 | 651.72 |
| peak MEM, Gb | 19.09 | 135.57 | 135.50 | 27.78 |

with an N50 value of 11.59 Mb. We also tested three other SMS assemblers (Canu, Flye, Ra) using this dataset. SMARTdenovo was superior to Flye and Ra in both total length and contig N50, but was inferior to Canu (Table 1).

The genome size of *D. melanogaster* is usually estimated to be 180 Mb. Of this, 60 Mb of the genome comprises centric heterochromatin, making it intractable for assembly [26]. Compared with the released reference [27], SMARTdenovo and Canu were able to create longer assemblies: ~146.29 Mb and ~161.97 Mb, respectively. SMARTdenovo and Canu performed better than the other two SMS assemblers, not only by creating longer assembly lengths, but also having higher coverage when aligned to the reference genome.

## Assembling the genome of the wild tomato

We also compared the performance of SMARTdenovo with that of three other SMS assemblers: Flye, Canu, and Ra using the dataset for the wild tomato *Solanum pennellii*. A total of 27.5 Gb Oxford Nanopore reads was downloaded from the European Nucleotide Archive database (PRJEB19787) [28]. A *k*-mer analysis of this dataset indicated that this accession of *S. pennellii* (LYC1722) has a genome size between 1 and 1.2 Gb [20]. We assembled a 30-fold Oxford Nanopore dataset and achieved an assembly of 902.96 Mb, with an N50 value of 339 kb (Table 2). We also tried Flye and Ra on this dataset: Flye obtained the longest genome sequence (1.27 Gb) and a higher N50 value (429 kb). Ra was unable to achieve the same sequence length as SMARTdenovo. When taking into account the computation time, SMARTdenovo required 651 central processing unit (CPU) hours, which was 70 hours faster than Flye (Table 2).

## DISCUSSION

SMARTdenovo is an accurate and efficient SMS assembler compatible with data formats output of both PacBio and Oxford Nanopore technologies. It comprises several command line tools: wtzmo, to overlap reads; wtclp, to trim low-quality regions and chimeras: wtlay, to generate the assembly graph layout; and wtcns to calculate the consensus. Based on the results of tests on the wild tomato dataset, we found that SMARTdenovo was more memory-intensive than the other SMS assemblers, but its performance was comparable, and it was faster. SMARTdenovo has been successfully used to assemble data from various species such as plasmids [29], protists [17, 30], fungi [19, 31, 32], microorganisms [33], and complex plants [18, 20, 21].

In addition to its solid performance, SMARTdenovo includes multiple algorithms that can be—and have been—useful for improving other programs. These algorithms have had a positive impact on popular SMS assemblers. For example, in developing SMARTdenovo, we introduced the first algorithm to use HPC-based *k*-mers, and this has now been incorporated into many other assemblers [10, 12, 22]. SMARTdenovo has had a more extensive impact on our development of the assembler Wtdbg2, as it includes several of its algorithms for handling long reads, including those for indexing, seed chaining, trimming, consensus, and some of the data formation.

There are several other algorithms within SMARTdenovo that have not yet been taken advantage of for use in other programs. An example includes its weighting algorithm for handling repeat regions, which significantly improves both its speed and the accuracy of the alignment. At this point, no other long read assemblers have this feature.

SMARTdenovo has been available on GitHub since 2015, but its performance not only remains comparable with current assemblers, it also has several advantages as described. Furthermore, given its excellent performance for use on corrected long sequence reads, it continues to be widely used in for genome assembly projects today [34–43].

## AVAILABILITY OF SOURCE CODE AND REQUIREMENTS

- Project name: SMARTdenovo
- Project home page: https://github.com/ruanjue/smartdenovo
- Operating systems: 64-bit Linux
- Programming language: C 93.3%, C++ 4.6%, Perl 1.5%, other 0.6%
- Other requirements: None
- License: GNU GPL-3.0
- RRID: RRID:SCR_017622

## DATA AVAILABILITY

A Code Ocean capsule to execute SMARTdenovo is available (Figure 2) [44]. Supporting data are available in the GigaScience GigaDB repository [45].

## DECLARATIONS
## LIST OF ABBREVIATIONS

HPC, homopolymer compressed; indel, insertion and deletion; OLC, overlap-layout-consensus; PacBio, Pacific Biosciences; SMS, single-molecule sequencing.

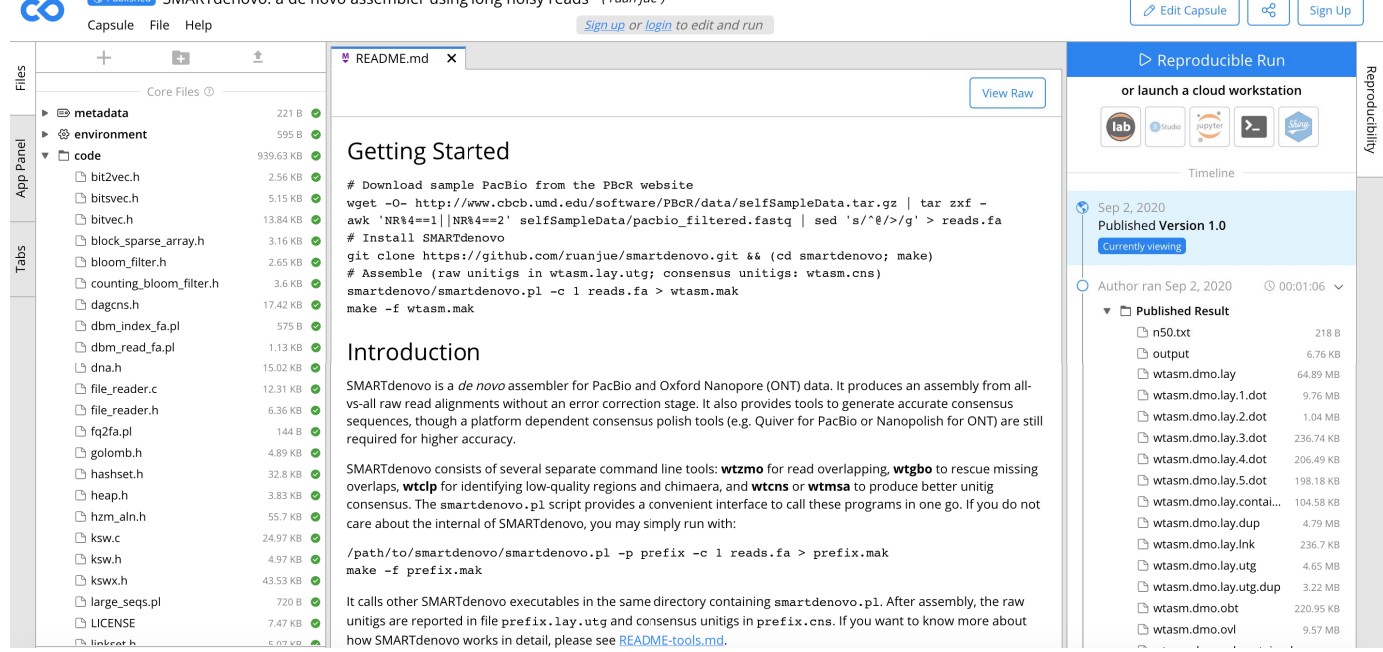

**Figure 2.** Code Ocean capsule to execute SMARTdenovo [44]. https://doi.org/10.24433/CO.4665826.v1

## ETHICAL APPROVAL
Not applicable.

## COMPETING INTERESTS
The authors declare that they have no competing interests.

## FUNDING
This study was supported by the National Key R&D Program of China (2019YFA0707003), the Natural Science Foundation of China (31822029).

## AUTHORS' CONTRIBUTIONS
J.R. initiated the program, coordinated the project. H.L.L. and J.R. wrote the manuscript. S.G.W., A.L.L and H.L.L conducted the software testing. J.R. and H.L.L revised the manuscript. All authors read and approved the final manuscript.

## ACKNOWLEDGEMENTS
Not applicable.

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
