## [Reviewer Report]

Reviewer name and names of any other individual's who aided in reviewerTrevor PesoutDo you understand and agree to our policy of having open and named reviews, and having your review included with the published manuscript. (If no, please inform the editor that you cannot review this manuscript.)YesIs the language of sufficient quality?YesPlease add additional comments on language quality to clarify if neededIs there a clear statement of need explaining what problems the software is designed to solve and who the target audience is? YesAdditional CommentsIs the source code available, and has an appropriate Open Source Initiative license <a href="https://opensource.org/licenses" target="_blank">(https://opensource.org/licenses)</a> been assigned to the code?NoAdditional CommentsThere is no license in the github repository.As Open Source Software are there guidelines on how to contribute, report issues or seek support on the code?YesAdditional CommentsGithub provides issue tracking, which can be used to report issues or seek support.Is the code executable?YesAdditional CommentsI needed to refer to their issues to find the modifications to the makefile necessary to compile the code.Is installation/deployment sufficiently outlined in the paper and documentation, and does it proceed as outlined?YesAdditional CommentsIs the documentation provided clear and user friendly?YesAdditional CommentsSparse but sufficient.Is there a clearly-stated list of dependencies, and is the core functionality of the software documented to a satisfactory level?YesAdditional CommentsHave any claims of performance been sufficiently tested and compared to other commonly-used packages? NoAdditional CommentsIn their evaluation of the fruit fly, they do not describe how misassembled_contigs and total_aligned_length_mb were determined. Total misassembly count would be a better metric of structural integrity in the assemblies than misassembled contig count. This is especially so as total number of contigs generated by the assemblers vary significantly, and the ratio of misassembled contigs to total contigs is worst for smartdenovo. There is also not any description of base-level accuracy, which should be included especially as they have a reference sequence. The runtime statistics should be presented for this sample.

In their evaluation of the wild tomato (Table 2), they include hours per CPU as a metric. Total CPU hours (num_cpu x total_runtime) is the correct metric; this is likely just mislabeled. No assembly quality metrics (such as were present in the fruit fly evaluation) were presented for this sample; I don't find the size and runtime statistics alone strong enough evidence of a high-quality assembly using ONT reads. Are there (ideally real world) examples demonstrating use of the software? YesAdditional CommentsIs automated testing used or are there manual steps described so that the functionality of the software can be verified?YesAdditional CommentsAny Additional Overall Comments to the AuthorIn the Introduction at L55 the authors do not cite Shasta as a current high-quality SMS assembler; I believe it should be included, especially as they refer to it in the methods at L78 and in the discussion at L211.

In the Discussion at L207, the authors claim to have introduced novel algorithms which "have been useful for improving other programs." I think this claim should be supported by more examples, especially as they don't describe any algorithms shared between themselves and MECAT or Flye. 

In Table 2, "peek MEM/Gb" should be "peak".RecommendationMinor Revisions

---

## [Reviewer Report]

Reviewer name and names of any other individual's who aided in reviewerRobert ReidDo you understand and agree to our policy of having open and named reviews, and having your review included with the published manuscript. (If no, please inform the editor that you cannot review this manuscript.)YesIs the language of sufficient quality?YesPlease add additional comments on language quality to clarify if neededNAIs there a clear statement of need explaining what problems the software is designed to solve and who the target audience is? YesAdditional CommentsIs the source code available, and has an appropriate Open Source Initiative license <a href="https://opensource.org/licenses" target="_blank">(https://opensource.org/licenses)</a> been assigned to the code?YesAdditional CommentsLicense: GNU GPL-3.0As Open Source Software are there guidelines on how to contribute, report issues or seek support on the code?YesAdditional CommentsBeing a github repo, forking, copying and contributing is all possible.Is the code executable?YesAdditional CommentsTested this tool out on 2 complex echinoderm species with genome sizes over 1GB each.Is installation/deployment sufficiently outlined in the paper and documentation, and does it proceed as outlined?YesAdditional CommentsFollowing the get started section on the GitHub page, I was able to install SMARTdenovo in less than 5 minutes in a linux environment.Is the documentation provided clear and user friendly?YesAdditional CommentsIs there a clearly-stated list of dependencies, and is the core functionality of the software documented to a satisfactory level?YesAdditional CommentsNo dependancies that I am aware of but did not test in a clean environment. I could easily have had all the pieces already, as I first installed this tool in 2017.Have any claims of performance been sufficiently tested and compared to other commonly-used packages? YesAdditional CommentsAre there (ideally real world) examples demonstrating use of the software? YesAdditional CommentsThey test a publicly available dataset from drosophila, D. melanogaster (SRX499318) and use that as a benchmark for performance.Is automated testing used or are there manual steps described so that the functionality of the software can be verified?YesAdditional CommentsThe tests here can easily be repeated by anyone.Any Additional Overall Comments to the AuthorThis is decent assembly tool that is straightforward and usable. It is not the tool for every de novo assembly out there but it is capable of producing comparable results to many other tools.
The figure has a few poor font resolutions that be improved. The spacing in the figure could be made better. The Figure caption could use more clarity on explaining the 4 images. RecommendationAccept

---

## [Reviewer Report]

Reviewer name and names of any other individual's who aided in reviewerErgude BaoDo you understand and agree to our policy of having open and named reviews, and having your review included with the published manuscript. (If no, please inform the editor that you cannot review this manuscript.)YesIs the language of sufficient quality?YesPlease add additional comments on language quality to clarify if neededIs there a clear statement of need explaining what problems the software is designed to solve and who the target audience is? YesAdditional CommentsIs the source code available, and has an appropriate Open Source Initiative license <a href="https://opensource.org/licenses" target="_blank">(https://opensource.org/licenses)</a> been assigned to the code?YesAdditional CommentsAs Open Source Software are there guidelines on how to contribute, report issues or seek support on the code?YesAdditional CommentsIs the code executable?YesAdditional CommentsIs installation/deployment sufficiently outlined in the paper and documentation, and does it proceed as outlined?YesAdditional CommentsIs the documentation provided clear and user friendly?YesAdditional CommentsIs there a clearly-stated list of dependencies, and is the core functionality of the software documented to a satisfactory level?YesAdditional CommentsHave any claims of performance been sufficiently tested and compared to other commonly-used packages? YesAdditional CommentsAre there (ideally real world) examples demonstrating use of the software? YesAdditional CommentsIs automated testing used or are there manual steps described so that the functionality of the software can be verified?YesAdditional CommentsAny Additional Overall Comments to the AuthorThis manuscript introduces SMARTdenovo, a de novo assembler for long noisy reads. The assembler has been published since 2015 widely used in various sequencing projects, and has had positive impacts on many de novo assemblers such as MECAT, Minimap2 and Wtdbg2. This manuscript details the algorithm step by step, including an overlapper wtzmo, a trimmer wtclp, a layout algorithm wtlay and a consensus algorithm wtcns. Novel ideas include homopolymer contracting and repeat weighing strategies. Besides the already published experimental results, this manuscript also exhibits some additional ones on species D. melanogaster and S. pennellii. SMARTdenovo can generate continuous and accurate contigs with small running time.

This manuscript is well-written and considering SMARTdenovo’s impact, I do not have any additional comment. Indeed, I have been waiting for the publication of SMARTdenovo for a long time, and it is honorable for me to have this chance of review.

Ergude Bao
School of Software Engineering
Beijing Jiaotong UniversityRecommendationAccept